# Heart Rate Variability as a Translational Dynamic Biomarker of Altered Autonomic Function in Health and Psychiatric Disease

**DOI:** 10.3390/biomedicines11061591

**Published:** 2023-05-30

**Authors:** Agorastos Agorastos, Alessandra C. Mansueto, Torben Hager, Eleni Pappi, Angeliki Gardikioti, Oliver Stiedl

**Affiliations:** 1II. Department of Psychiatry, School of Medicine, Faculty of Health Sciences, Aristotle University of Thessaloniki, 56430 Thessaloniki, Greece; 2Center for Neurogenomics and Cognitive Research, Vrije Universiteit (VU) Amsterdam, 1081 HV Amsterdam, The Netherlands; a.c.mansueto@uva.nl (A.C.M.); torben.hager@gmail.com (T.H.);; 3Centre for Urban Mental Health, University of Amsterdam, 1081 HV Amsterdam, The Netherlands; 4Faculty of Health Sciences, School of Medicine, Aristotle University of Thessaloniki, 54124 Thessaloniki, Greece; elenipappi@auth.gr (E.P.); aggardik@auth.gr (A.G.); 5Department of Health, Safety and Environment, Vrije Universiteit (VU) Amsterdam, 1081 HZ Amsterdam, The Netherlands

**Keywords:** heart rate variability, autonomic nervous system, central autonomic network, sympathetic nervous system, parasympathetic nervous system, vagus, heart, time domain, frequency domain, nonlinear measures, biomarkers, neurophysiology

## Abstract

The autonomic nervous system (ANS) is responsible for the precise regulation of tissue functions and organs and, thus, is crucial for optimal stress reactivity, adaptive responses and health in basic and challenged states (survival). The fine-tuning of central ANS activity relies on the internal central autonomic regulation system of the central autonomic network (CAN), while the peripheral activity relies mainly on the two main and interdependent peripheral ANS tracts, the sympathetic nervous system (SNS) and the parasympathetic nervous system (PNS). In disease, autonomic imbalance is associated with decreased dynamic adaptability and increased morbidity and mortality. Acute or prolonged autonomic dysregulation, as observed in stress-related disorders, affects CAN core centers, thereby altering downstream peripheral ANS function. One of the best established and most widely used non-invasive methods for the quantitative assessment of ANS activity is the computerized analysis of heart rate variability (HRV). HRV, which is determined by different methods from those used to determine the fluctuation of instantaneous heart rate (HR), has been used in many studies as a powerful index of autonomic (re)activity and an indicator of cardiac risk and ageing. Psychiatric patients regularly show altered autonomic function with increased HR, reduced HRV and blunted diurnal/circadian changes compared to the healthy state. The aim of this article is to provide basic knowledge on ANS function and (re)activity assessment and, thus, to support a much broader use of HRV as a valid, transdiagnostic and fully translational dynamic biomarker of stress system sensitivity and vulnerability to stress-related disorders in neuroscience research and clinical psychiatric practice. In particular, we review the functional levels of central and peripheral ANS control, the main neurobiophysiologic theoretical models (e.g., polyvagal theory, neurovisceral integration model), the precise autonomic influence on cardiac function and the definition and main aspects of HRV and its different measures (i.e., time, frequency and nonlinear domains). We also provide recommendations for the proper use of electrocardiogram recordings for HRV assessment in clinical and research settings and highlight pathophysiological, clinical and research implications for a better functional understanding of the neural and molecular mechanisms underlying healthy and malfunctioning brain–heart interactions in individual stress reactivity and psychiatric disorders.

## 1. Introduction

Assessing autonomic (re)activity is vital for the affective, cognitive and physical neuroscientific field, and should not only be used as a theoretical ground for further research, but also as a starting point for clinical applications. Although autonomic imbalance can be easily measured, these possibilities are, however, generally used sparsely by clinicians. The aim of this article is to provide basic knowledge on autonomic function and assessment and to support a much broader use of heart rate variability (HRV) as a valid, transdiagnostic and fully translational (applied into clinical applications) dynamic biomarker of stress system sensitivity and vulnerability to stress-related disorders in research and clinical practice. Here, we review the functional levels of central and peripheral autonomic control, the main underlying neurobiophysiologic theoretical models, the pathways of autonomic influence on cardiac function and the definition and main aspects of HRV and its different assessment measures (i.e., time, frequency and nonlinear domains). We also provide recommendations for the proper use of electrocardiogram recordings for HRV assessment in clinical and research settings and highlight pathophysiological, clinical and research implications for a better functional understanding of the neural and molecular mechanisms underlying healthy and malfunctioning brain-heart interaction in individual stress reactivity and psychiatric disorders.

### 1.1. Stress and the Stress System

Stress is defined as a state of threatened homeodynamic balance [1,2] in an organism in response to a wide range of intrinsic or extrinsic, objective or subjective challenges or stimuli, defined as stressors [3,4]. The human stress system is a highly sophisticated system with central and peripheral components, which aids the self-regulation and adaptability of the organism by redirecting energy according to the organism’s needs at any given time [1,2,5,6]. The central greatly interconnected brain areas of this system are located in the hypothalamus and the brainstem. They interact with several other major brain nuclei and neuromodulatory systems, while being affected by numerous cognitive, emotional, neurosensory, humoral, immune, homeostatic and peripheral somatic signals through different pathways [1,2,6,7]. The peripheral components of the stress system include the hypothalamic–pituitary–adrenal (HPA) axis, the sympathetic adrenomedullary (SAM) system and the autonomic nervous system (ANS). In view of its negative connotation, the term ‘stress’ should be reserved for conditions in which clear maladaptive (pathological) responses are observed. This needs to be clearly discriminated from adaptive responses to short-lasting challenging conditions that occur in any mammal including humans.

### 1.2. The Autonomic Nervous System (ANS)

The ANS, although not under overt voluntary control (autonomous), plays a crucial role in the maintenance of homeodynamic balance by providing rapid and precise regulation of organ and tissue functions such as the respiratory, cardiovascular, digestive, endocrine systems and many other systems, and by adjusting to ongoing physical, emotional and cognitive challenges, resting state and sleep [8,9]. On the contrary, an autonomic imbalance is associated with—decreased dynamic adaptability, increased morbidity and mortality of an organism [9,10,11,12]. Although the ANS is one of the larger control systems, it has been traditionally divided into three, or sometimes four, divisions (tracts): the sympathetic nervous system (SNS), the parasympathetic nervous system (PNS), the sympathetic adrenomedullary (SAM) system and, the largest one, the enteric autonomic division [13,14]. In particular, the numerous multi-level and bidirectional interactions between the PNS and the SNS provide a joint fine-tuning of peripheral autonomic functions according to physiological demands, and also secure system lability and variability. SNS activation generally prepares the body for physically strenuous or emotionally challenging conditions (fight-or-flight response). Parasympathetic (vagal) activity, on the other hand, predominates during resting conditions and in connection with energy conservation or the regulation of basic body functions (e.g., digestion, defecation, urination) due to its tonic function and is, therefore, particularly implicated in the pathophysiology of cardiovascular diseases and other comorbidities [10,15]. However, the focus on the different SNS and PNS actions has led to the crucial misconception that the two divisions are somehow functionally opposed to each other. On the contrary, the SNS and the PNS act in concert through a dynamic balanced interdependent state that involves coupled and uncoupled (independent) activation modes that act in different time scales. For example, due to its tonic function, the PNS can both assist and antagonize SNS functions by withdrawal under resting state conditions or increasing its activity (frequency of neuronal discharge), respectively, as observed during sleep [11,12,16,17].

### 1.3. Central Autonomic Network (CAN)

The central autonomic network (CAN) mediates the task- and division-specific regulatory control of central pathways for peripheral autonomic regulation [18]. The CAN includes the insular cortex, central nucleus of the amygdala, hypothalamus, periaqueductal gray matter, parabrachial complex, nucleus tractus solitarius (NTS) and ventrolateral medulla (VLM) [19,20]. High-order autonomic control associated with cognitive and emotional functioning is mediated by the insular cortex and the amygdala through hypothalamic–brainstem pathways [9]. A network of respiratory, cardiovagal and vasomotor neurons is contained in the NTS, periventricular nucleus (PVN) and VLM regions and receives afferent vagal sensory input from thoracic and abdominal viscera and other cranial nerves. These structures, accordingly, modulate the activity of the preganglionic autonomic neurons. It is very important to note that central autonomic modulation by the CAN does not simply rely on a monolithic network of brain regions, but instead features certain task and division specificity with several reciprocal interconnections, leading to a high neurochemical complexity according to its state-dependent activity [8,9,18,21,22]. CAN dysregulation may affect downstream autonomic output (brainstem) centers, thereby altering peripheral ANS activity and, eventually, the dynamics of different subordinate systems (e.g., cardiovascular system) [22,23,24,25,26] (Figure 1). Accordingly, CAN dysregulation may be critically involved in psychiatric disorders, essential hypertension, obesity and other medical conditions [27].

## 2. The Autonomic Nervous System (ANS) and the Heart

One of the most appropriate systems for the functional assessment of ANS activity is the cardiovascular system, as responses of the effector tissues are prompt, easy to access and have a high clinical and prognostic value [28,29]. The close functional connection between the brain and the heart was described by the French physiologist Claude Bernard over 150 years ago [26]. Commenting on his work, Charles Darwin further proposed that heart and brain were vagally linked, which was an astonishing statement at the time it was made [30]. The interaction between these two highly important organs in the body is critically associated with normative neurodevelopment and aging [31]. Emotionally or physically driven stress-related changes in brain–heart interaction may affect healthy development and represent a vulnerability to stress and a risk factor for all-cause morbidity and mortality.

The heart features a number of different effector tissues: the sinoatrial (SA) node, the atrioventricular (AV) node, the atrial and the ventricular myocardium [32]. Modulation of ion channels, particularly in the primary and secondary pacemaker (SA and AV node, respectively), influences heart function in three distinct, but interdependent, ways: chronotropy, inotropy and dromotropy. This modulation is predominately influenced by the ANS through the tonic drives of, and interactions with, both sympathetic and parasympathetic cardiac nerves in an asymmetric manner depending on demand. Right-sided SNS and PNS fibers reach the SA node and modulate particularly chronotropic activity (i.e., influencing frequency of contraction; SNS: positive chronotropy; PNS: negative chronotropy), while left-sided SNS and PNS fibers innervate the AV node and myocardium, and thereby modulate dromotropic (i.e., influencing the electrical impulse conduction between the SA and AV nodes (PQ interval of the human ECG)) and inotropic (i.e., influencing force of contraction) activity in the same manner [33,34,35] (Figure 1).

Resting heart activity is under constant tonic inhibitory control by the PNS and its dominance over the SNS [36,37]. In addition, heart rates (HR) are characterized by beat-to-beat variability over a wide range, implicating vagal dominance as SNS cardiac influence is too slow and long-lasting to produce rapid beat-to-beat changes (e.g., respiratory sinus arrhythmia) [38,39]. Thus, simultaneous PNS and SNS cardiac co-activation leads to more efficient cardiac function than SNS activation alone as it permits both longer ventricular filling and stronger myocardial contraction [40]. In contrast, autonomic imbalance leads to reduced dynamic flexibility and functional complexity and, subsequently, to increased vulnerability to pathologies and compromised cardiac health. In particular, the PNS, through its tonic inhibitory influence on the heart, is heavily implicated in the pathophysiology of cardiovascular diseases and other comorbidities [10].

## 3. Heart Rate Variability (HRV)

Although the cardiovascular system is particularly appropriate for the functional assessment of ANS activity, standard cardiac measures (e.g., HR) offer only limited information on ANS activity, as short- and long-term HR variations are not considered. One of the best established and most widely used non-invasive methods for the quantitative assessment of ANS activity is the computerized analysis of heart rate variability (HRV) [41,42]. HRV reflects heart–brain interactions and ANS dynamics as it results from HR oscillations within its physiological range (beat-to-beat variability) and is controlled and modulated by autonomic modulation of intrinsic cardiac pacemakers. Higher HRV is related to parasympathetic dominance, while low HRV is related to enhanced sympathetic and/or attenuated parasympathetic cardiac modulation for mobilization of energy resources, particularly during high attention, arousal and stress. In general, HR and HRV are inversely related in physiological states, i.e., the higher the HR, the lower the HRV. Thus, strictly speaking, HR and HRV are not independent measures. Generally, a reduced HRV is correlated with increased cardiovascular mortality [43]. However, as observed in mouse models after systemic 5-HT_1A_ receptor activation [44], an increased HRV index of enhanced sympatho-vagal antagonism] also indicates a pathological state. 5-HT_1A_ receptor overactivation in mice copies the cardiovascular effects of selective serotonin reuptake inhibitors (SSRIs) overdosing in humans as described in the FDA recommendation on the SSRI Citalopram (see http://www.fda.gov/drugs/drugsafety/ucm297391.htm (accessed on 10 April 2023)).

## 4. Heart Rate Variability Measures

Although HRV is a useful biomarker, there is no common agreement on which of the many available HRV measures to use in research and clinical practice. HRV measures can mainly be divided in two main categories: linear and nonlinear measures (Table 1).

### 4.1. Linear Measures

Using linear algorithms, HRV can be analyzed in time or frequency domain. Standard time and frequency domain parameters were defined by the 1996 Task Force on HRV and are the most commonly used HRV measures.

#### 4.1.1. Time Domain Measures

Time domain indexes are the first measures applied and are a basic way to calculate HRV as they are based on the normalized time between two detected heartbeat QRS events (NN interval, excluding unreliable intervals). Time domain HRV indexes are strictly correlated with each other and are normally calculated as standard deviation of the NN intervals (SDNN), the root mean square of subsequent interval differences (RMSSD), and the percentage of the number of pairs of adjacent NN intervals that differ in length by >50 ms (NN50%). Low SDNN, NN50 and RMSSD denote low vagal tone [41].

#### 4.1.2. Frequency Domain Measures

*Frequency domain indexes* are more elaborated measures based on spectral analysis (fast Fourier transformation) and are mostly used to assess HRV. Frequency measures represent the power (energy) as a function of frequency with regard to specific frequency bands commonly based on fast Fourier transformation [41]. HRV in the frequency domain is normally calculated by analysis of the power of two distinct frequency bands (low frequency (LF) 0.04–0.15 Hz and high frequency (HF) 0.15–0.4 Hz). The results are frequently presented as a power percentage (%) of each frequency band relative to the total power or in normalized units with the LF/HF ratio as a potential measure of sympatho–vagal balance. Calculating the percentage (%) of frequency components can minimize the effect of total power changes on the LF and HF component and avoid false interpretation of the spectral components in absolute units [41]. Efferent vagal activity is suggested to be the major contributor to the HF component, although it may also be affected by sympathetic activation by up to 10% [41,47]. The interpretation of the LF component is more controversial. While earlier studies suggested LF as an indicator of sympathetic activity, recent research has argued that LF may instead reflect a complex mix of sympathetic, vagal and other unidentified factors, with vagal activity accounting for the largest influence on the variability in this frequency range [47,48]. This also challenges the hypothesis that the LF/HF ratio accurately mirrors sympatho-vagal balance [47,48]. Another very important translational disadvantage is the fact that the frequency bands used in animal models differ from human frequency bands. Finally, frequency bands represent only a fraction of the whole energy spectrum of the power analysis, thus introducing bias into the interpretation of relative results.

#### 4.1.3. Geometrical Measures

A third, less frequently used form of analysis is the Poincaré plot analysis. The Poincaré plot of RR intervals, commonly described as a nonlinear measure, is not described by indexes that reflect nonlinear features of HRV [49] but shows an extremely high correlation with time domain measures such as RMSSD [50]. Therefore, Poincaré plot analysis should more appropriately be referred to as a geometrical time domain measure. It is a graphical representation obtained by plotting the durations of the preceding RR intervals against the durations of the current RR intervals, i.e., RR_n+1_ is represented on the *Y*-axis, RR_n_ on the *X*-axis. Various parameters describe the Poincaré plot, with SD1 and SD2 being the most commonly used. SD1 is the dispersion (standard deviation) of the distribution of points perpendicular to the line of identity, and is a measure of short-term HRV. SD2 is the dispersion of the cloud of points along the line of identity, and is a measure of long-term HRV.

### 4.2. Nonlinear Measures

Two major functional properties of HR dynamics, non-stationarity (drift-like behavior of HR) and interdependence (correlation of heartbeat intervals in a temporal sequence under physiological conditions), formally require the assessment of HR dynamics by non-standard methods to determine the organization of any apparently ‘noisy’ HR signal with its internal correlation and level of complexity [17]. Nonlinear (fractal) indexes were introduced relatively recently (in the 1990s) as methods to measure HRV. These methods are not affected by the non-stationarity of heartbeat interval fluctuations and also determine the complex temporal interactions of hemodynamic, electrophysiological and humoral variables as well as the autonomic and central nervous system (CNS) regulation which affects HR dynamics [17]. They include power law exponents, different entropy measures and, most importantly, detrended fluctuation analysis (DFA) [17]. Deriving from the random walk theory, DFA indicates ANS dysregulation by using scaling coefficients (α) as a measure of the correlation of heartbeats in their temporal sequence and is a powerful tool for the characterization of various complex systems as the DFA scaling coefficient is resistant to HR changes due to altered physical activity [51]. As tonic vagal activity secures long-range correlation of heartbeat interval fluctuations in a healthy state [52], the DFA scaling coefficient is, thus, considered a sensitive readout to assess functional alterations indicative of reduced vagal tone. A scaling coefficient *α* = 1.0, indicating the persistent long-range correlation of successive heartbeat intervals, is observed in all physiological HR dynamics across mammalian species (incl. mice and man). Pathological states are characterized by a breakdown of long-range correlations, with an α strongly deviating from 1.0. An *α* = 1.5 reflects ‘Brownian noise’ and only indicates a short-term correlation between one heartbeat interval and the next one, and is observed as a blocked tonic parasympathetic outflow to the heart (i.e., atropine treatment, heart transplantation) [52]. An *α* = 0.5 indicates ‘white noise’ and a randomness by showing a lack of any correlation between heartbeat intervals. A breakpoint in the straight-line relationship separates slopes of coefficients into different distinct ranges, *α*_fast_ and *α*_slow_ (i.e., short-range and long-range scaling) [17].

Given that nonlinear analyses are scale-invariant, they can provide functional information about the dynamics of heartbeat interval fluctuations which can be used translationally between animal models and humans (i.e., irrespectively of species-specific HR differences) [53]. In addition, nonlinear methods have been repeatedly shown to be highly sensitive predictors of ANS dysregulation-related cardiac dysfunction even in the absence of cardiac disease, suggesting a primary role of CNS dysfunction in the elevated cardiac risk seen in affective and stress-related disorders [17,54]. Nevertheless, to date, this method is still only applied in a relatively limited number of publications due to its complexity [55].

Finally, multifractal [17,52,56] and entropy-based [50,57] measures are even more sensitive than unifractal (or monofractal) measures in identifying pathological states. However, due to their complexity and the large spectrum of different entropy-based approaches, nonlinear approaches are less common as standardized measures in the analysis of HRV or better HR dynamics.

## 5. Recommendations for HRV-Relevant ECG Recordings

One of the most important issues is how to record ECG in a standardized manner [41], for example by using medically approved and certified ECG holter recording systems and digital recording systems with a suitable sampling rate (≥250 Hz). This also includes recordings at the same time of day (i.e., to exclude diurnal differences) and in comparable physiological (e.g., activity) and psychological (e.g., stress-challenge) states. Although several studies have applied such standardized assessments in the past, most analyses only include ECG recordings of predominantly short time periods (5–30 min). Such shorter intervals are appropriate to provide time and frequency domain measures, but are insufficient or suboptimal to assess HR dynamics by non-linear methods [17]. Non-linear methods require much longer ECG/HR recordings, while ECG/HR signals with transient data loss need to be interpolated when only a few beats are missing and ECG which is heavily confounded by artifacts needs manual editing for nonlinear analysis, if usable at all. Thus, nonlinear analysis depends on high-quality, long-term ECG/HR data.

Another important precondition for HRV analyses is proper quality control for detection of the R-peaks of the ECG signal. Compared to processed raw data, commercial systems often provide only suboptimal ECG detection algorithms. These algorithms may only negligibly affect HR analyses and are not sensitive enough for time domain measures such as RMSSD or nonlinear measures. Consequently, a combination of time-consuming manual and computerized editing by experienced scientists is obligatory before actual data analysis is performed. Last but not least, studies investigating HRV measures should always control for received medication which may affect HRV and should account for a range of important factors and/or exclusion criteria (e.g., presence or history of cardiovascular diseases, or abuse/dependency on alcohol or other substances), that may otherwise confound the results and conclusions. A diagram depicting the individual steps from ECG recording to data analysis is shown in Figure 2.

## 6. Pathophysiological Implications

In human research, HRV reflects individual capacity for physiological and emotional regulation. In relation to this, recent studies have suggested HRV as a biomarker to monitor emotional regulation in association with neurophysiological changes in several psychiatric disorders, including depression and anxiety disorders.

Normally, as integral components of the CAN, the PFC and amygdala mediate high-order autonomic control associated with cognitive perception and emotional responses, respectively, through the vagus nerve [10] and hypothalamic–brainstem pathways [9]. PFC hypofunction, as observed, for example, in stress-related disorders such as PTSD, leads to the deficient cognitive influence of emotional responses and subsequently to exaggerated amygdala activity [58,59]. PFC hypofunction may, thus, affect downstream autonomic core centers (e.g., hypothalamus, brainstem) and influence the peripheral ANS activity and dynamics of heartbeat intervals, resulting in a decreased HRV [22,23,24,25,26]. Accordingly, PFC hypoactivity is reflected in lower HRV and in a related reduced capacity for emotional regulation, decreased psychological flexibility and defective social engagement [58,59].

Low HRV is associated with higher overall mortality, specifically heart mortality (e.g., heart infraction), and is considered a valid marker of heart disease [10,20,43]. Accordingly, CAN dysregulation may be involved in several mental health disorders and chronic non-communicable diseases (e.g., hypertension, obesity, etc.) [27] and may also be at the core of sudden unexpected death in epilepsy (SUDEP) as indicated by concomitant ECG and EEG recordings [60].

The high comorbidity of stress-related psychiatric disorders and cardiovascular disease [61,62,63,64,65,66] underlines the important pathophysiological link between these disorders and compromised neuroautonomic control [67,68,69]. Particularly, the elevated risk of cardiovascular failure after emotionally challenging events [70] indicates the crucial role of stress-induced adjustments made by the sympathetic–adrenal medullary system and demonstrates the modulating effects of emotional challenges for the emergence of cardiac risk and fatal outcomes as reported in epidemiological studies [71]. In addition, altered autonomic reactivity also shows significant age-related changes (e.g., diminished autonomic reactivity and poorly coordinated autonomic discharge), leading to an impaired ability to adapt to environmental or intrinsic visceral stimuli in the elderly [72].

A variety of studies, systematic reviews and meta-analyses have repeatedly shown that HRV displays good validity as a biomarker in psychiatry. Lower HRV and HRV reactivity has been associated with most psychiatric disorders (e.g., PTSD, anxiety disorders, depression, schizophrenia, bipolar disorder, autism), with higher symptom severity, as well as with specific endophenotypes [73,74,75,76,77,78,79]. Especially in depression, baseline HRV has been associated with a history of depression (i.e., recurrence) [80], severe symptoms and response to antidepressant treatment [81]. In particular, HRV has been used as a biomarker for treatment stratification in MDD, as it could differentiate between different depression phenotypes having different outcomes following antidepressant treatment [82]. HRV has additionally been proven to be a valid biomarker in the prediction of early cognitive impairment in healthy populations [83], proximal suicide risk in psychiatric patients [84], language skills in children with autism [85] and treatment response in OCD [86].

## 7. Neurobiophysiological Theoretical Models

The Johns Hopkins Tripartite Model of Resistance, Resilience and Recovery suggests that individual ability for resistance to stress, rapid and effective rebound from stress and functional improvement after stress plays a crucial role in healthy adaptation, development of stress-related disorders and recovery treatment [87]. Research carried out during the past two decades suggests a correlation between autonomic reactivity, emotion regulation, cognitive function and pathology, with autonomic dysregulation having a major affect on overall mental and physical health and functioning [88]. Importantly, the neural circuits of autonomic regulation, attentional regulation and affective regulation that allow the organism to meet the challenges of an ever-changing environment overlap heavily. The two main conceptual–theoretical cornerstones of this perspective are Porges’ Polyvagal Theory and Thayer and Lane’s Neurovisceral Integration Model, which both implicate the PNS, in particular, in the regulation of the central stress system.

### 7.1. The Polyvagal Theory

According to Porges’ polyvagal theory [89,90], autonomic function is linked to behavior, with several neural circuits involved in the regulation of the autonomic state. In particular, Porges suggested that the vagal system contains specialized subsystems with competing roles that regulate adaptive responses, and which are crucial to the development of emotional experience and affective processes central to social behavior. Porges suggested that unmyelinated fibers from the dorsal motor complex are involved in regulating the “freeze response”, while myelinated nerves from the nucleus ambiguous represent a vagal brake which allows for self-regulation and ability to inhibit sympathetic outflow. His theory implies that a healthy vagal function of the ANS sets the limits or boundaries for the range of one’s emotional expression, quality of communication and ability to self-regulate emotions and behaviors. A standardized assessment of vagal tone could, thus, serve as a potential marker for one’s ability to self-regulate and could mirror different types or classes of behavior. Although broader interest has been shown in this theoretical model, strong experimental support is fairly limited, as reported in a review on HRV in autism spectrum disorder in children [91].

### 7.2. The Neurovisceral Integration Model

The Neurovisceral Integration Model is one of the most influential psychophysiological models addressing the interplay between CNS and somatic functioning (brain–body interaction). The model describes how a set of brain areas involved in cognitive, affective and autonomic regulation are related to HRV and cognitive performance [9]. Thereby, anatomical and functional interconnections of the CAN are suggested to link the brainstem with forebrain structures and the amygdala with the mPFC through feedback and feed-forward loops and to control visceromotor, neuroendocrine and behavioral responses critical for goal-directed behavior, adaptability and overall health. They further propose a central role for the vagus in adaptation to the environment, through providing negative feedback on sympatho–excitatory stress responses [10]. For example, vagally-mediated HRV is linked to higher-level executive functions and the ability to inhibit unwanted memories, and thereby reflects the functional capacity of the brain structures that support working memory and emotional and physiological self-regulation. Taken together, vagal inhibitory processes can be viewed as negative feedback circuits that allow for the interruption of ongoing behavior and the re-deployment of resources to other tasks and are, thus, central in the interplay of the cognitive, affective, behavioral and physiological concomitants of normal and pathological emotional states [9].

## 8. Clinical Implications

The incorporation of autonomic (re)activity into a broader perspective on emotions, mental health and good cognitive and physical functioning should not only be used as a theoretical ground for further research, but also as a starting point for clinical applications [88]. Nevertheless, although particularly autonomic imbalance can be easily measured and is also influenced by methods that are already available in primary care, these possibilities are generally overlooked by clinicians [92]. Simple strategies for autonomic function improvement and increasing cortical blood flow could be used to improve autonomic activity and reactivity [72]. For example, as physical activity influences both resting autonomic activity and autonomic reactivity, regular moderate aerobic exercise (e.g., walking) could have a positive effect on ANS reactivity [93].

Additional clinical intervention strategies could include more specific treatment alternatives, such as pharmacotherapy (e.g., SSRI) and somatic afferent stimulation (e.g., stroking skin, acupuncture, vagus nerve stimulation, HRV coherence training/biofeedback), to restore autonomic balance. Instead of exclusively targeting sympathetic activation as in the past, physicians should rather attempt to increase vagal tone. In particular, there has been increasing interest in treating a wide range of disorders with implanted pacemaker-like devices which stimulate the vagal afferent pathways for a broad range of diseases (e.g., obesity, depression, anxiety, epilepsy, migraine, chronic pain, etc., [60,94,95,96]). In addition, drugs affecting CAN activity (e.g., SSRIs) and circadian rhythm, substances reducing oxidative stress or inflammation, or influencing stress-system dysregulation effects in the periphery (e.g., glucocorticoid receptor modulators), or even metabolism-altering agents, have the potential to effectively disrupt the chronic vicious cycle of stress progression and its effects on the body.

## 9. Research Implications

HRV can serve as a transdiagnostic biomarker for cardiovascular risk and cognitive-emotional assessment. Given its fully translational properties, expanded data-related nonlinear analysis in particular could be an excellent diagnostic and potentially therapeutic tool in future research and may expand our functional understanding of the dysregulation of HR dynamics, as it bridges basic research and clinical research [53,57]. In addition, longitudinal assessments of HR dynamics before and after treatment would be of great value in determining precise changes in ANS reactivity and normalization (recovery). However, in view of the broad range of cardiovascular diseases that may also impact on HRV, autonomic markers must be combined with complementary markers for unambiguous identification, especially of early pathological states, in order for correct conclusions about precise psychopathology diagnosis to be drawn.

In addition, neuroautonomic responsiveness to stressors can be of the utmost importance since hyper- and hypo-responsiveness may be used as indicator for altered autonomic function in response to emotional challenge that may be affected by an altered baseline state. For example, challenge-induced response differences can be affected by serotonin transporter-linked polymorphic region genotype [97], long-term SSRI treatment [98], metabotropic glutamate signaling [99] and HPA-axis manipulation [100] in healthy controls, but also by the presence of depression history in patients [80]. This demonstrates that normal challenge-induced HR response magnitudes can be attenuated in psychopathology because of maladaptive changes (allostatic load) [101].

When assessing HRV, accounting for medications affecting HRV is crucial and is sometimes an important exclusion factor. For example, the results of both hydrophilic and lipophilic β-blockers on HRV suggest a relative complex, but rather positive, association [102,103], as they seem to improve the baroreflex control of HR leading to an increase in RMSSD, pNN50 and HF power with a decrease in the LF/HF ratio [104], possibly through vagal facilitation [105] and a more favorable sympathovagal balance. β-blockers are, accordingly, suggested to accelerate HRV recovery in acute heart disease (e.g., heart infraction) patients [106]. Concerning other drugs, such as antiarrhythmics (e.g., amiodarone), calcium channel blockers, antihypertensives, cardiac glycosides, or nitrates, no solid conclusion can be drawn because of the low number of human trials and the nature of patients included (mostly acute or chronic cardiac, or multimorbid patients) [102].

On the other hand, it is important to note that, since many studies report increased HR and reduced HRV (which are generally inversely related) in psychiatric disorders, b-adrenergic antagonists (β-blockers) have been suggested as adjunctive pharmacotherapy to successfully lower HR and increase HRV in anxiety but not in panic disorder and PTSD [107]. However, the focus on β-blockers such as propranolol has been more on fear memory extinction of traumatic memories than on the beneficial effects on autonomic function in psychiatric disorders [108]. Additionally, negative effects of propranolol on depression have been claimed but remain inconsistent [109]. Thus, a tradeoff of the effects of cardiovascular and psychiatric pharmacotherapies is likely, as is also apparent from the inconsistent effects of SSRIs and HRV [108].

Finally, the specific prognostic value of pre-exposure HR dynamics in the development of PTSD after trauma exposure is of crucial importance and is currently being pursued in prospective studies which suggest that an initially reduced HRV is a risk factor for PTSD [110]. Related research could be valuable for the identification, based on cardiovascular biomarkers, of susceptible individuals in high-risk professions (e.g., deployed soldiers) before potential traumatic exposure, and could enable innovative preventive strategies and psycho-chronobiological treatment possibilities in PTSD patients and high-risk trauma-exposed populations [111,112,113,114,115]. In addition, HR data collection in exposure settings could enhance our understanding of PTSD-related neuroautonomic alterations as a potential consequence of the altered fear circuitry affecting PFC function and emotional responses [116,117].

In this respect, future studies focusing on the diurnal variability of autonomic stress reactivity and its role in the development of stress-related disorders and cardiac comorbidity are especially needed. In addition, seriously challenging conditions, despite their ethical problems, should be explored more thoroughly, particularly with respect to HRV responses and recovery, as well as to determine any therapeutic efficacy. Therefore, studies should focus more on unmedicated patients and/or consider a range of important exclusion criteria and factors that may influence results and conclusions when investigating HRV measures in healthy individuals and disease cohorts.

## 10. Future Directions

The neurovisceral integration model proposes that altered stress system reactivity plays an important role in all major allostatic systems in the pathway from excessive stress to chronic disease [10]. The understanding of the risk factors leading to chronic stress system malfunction may yield important insights into the etiopathology, progression, prevention and treatment of the chronic, non-communicable diseases which are currently one of the most important and major public health concerns [118]. Especially when functional interactions between ANS reactivity and somatic subsystems are considered, diverse patterns of behavioral maladapation may be subsumed into a single disease model. In this model, autonomic imbalance represents the final common pathway leading to increased morbidity and mortality from a wide range of medical conditions, while the assessment of the autonomic imbalance may provide a unifying framework for investigating the impact of risk factors, including biological, behavioral, psychosocial and environmental factors on health and disease [26,43]. In particular, chronic activation of inhibitory cortico-subcortical circuits leading to low parasympathetic activation and prefrontal hypoactivity may structurally, as well as functionally, link altered psychological processes with health-related physiological consequences [119]. Thereby, measures of ANS reactivity can be viewed as an autonomic, transdiagnostic biomarker of self-regulation, cognitive control and overall state of health [120]. HRV, in particular, can be easily used to assess autonomic imbalance in the staging of chronic diseases and the classification of morbidity and mortality risk [43].

Therefore, HRV assessment, and especially more longitudinal monitoring by nonlinear methods, may improve our interpretations of autonomic dysregulation of HR dynamics and serve as a sensitive clinical biomarker which may even have potential prognostic value in future research into stress-related disorders. A full translation of findings on neuroautonomic heart control can only be achieved by nonlinear methods [121], as these are better suited to determining physiological from pathological changes that can be easily misinterpreted as beneficial due to enhanced sympatho-vagal antagonism with increased HR variability and blunted responsiveness to stress as a pathological state [44].

## 11. Conclusions

Individual differences in stress (re)activity may vitally affect adaptive responses and may possibly explain individual differences in stress resilience and the progression of stress-related disorders. Therefore, a functional understanding of the brain–heart interaction requires considerable improvement through preclinical and prospective clinical research, as the role of the specific brain areas involved in central neuroautonomic dysregulation in response to specific stressors and stress-related disorders is still largely unclear. The further identification of biological factors that affect stress reactivity is, thus, of major importance for connecting cognitive, emotional, psychosocial and environmental stress factors with inter-individual variation in disease outcome. Since emotional and arousal circuitry largely overlaps with the CAN, HRV, especially nonlinear HR measures, could serve as suitable and fully translational biomarkers for stress system sensitivity, vulnerability and individual risk, with a potential much broader use in research and clinical practice.

## Figures and Tables

**Figure 1 biomedicines-11-01591-f001:**
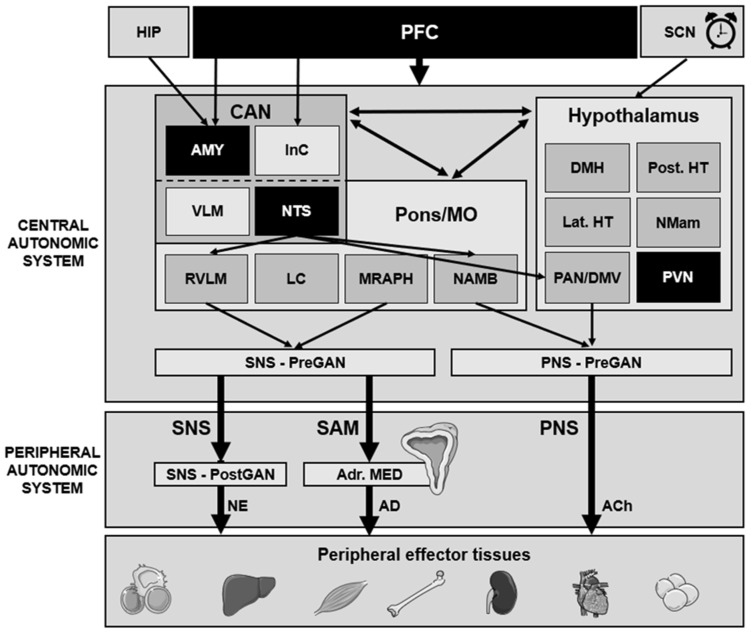
Simplified diagram of the central and peripheral autonomic nervous system network with major brain areas. Ach, acetylcholine; AD, adrenalin; Adr. MED, adrenal medulla; AMY, amygdala; CAN, central autonomic network; DMH, dorsomedial hypothalamus; DMV, dorsal motor nucleus of the vagus nerve; HIP, hippocampus; InC, insular cortex; Lat. HT, lateral hypothalamus; LC, locus coeruleus; PFC, prefrontal cortex; MO, medulla oblongata; MRAPH, median raphe nuclei; NAMB, nucleus ambiguous; NE, norepinephrine; NMam, nucleus mamillaris; NTS, nucleus of the solitary tract; PAN, pre-autonomic hypothalamic nucleus; PNS, parasympathetic nervous system; PreGAN, preganglionic autonomic neurons; Post. HT, posterior hypothalamus; PostGAN, postganlionic autonomic neurons; PVN, paraventricular nucleus; RVLM, rostral ventrolateral medulla; SAM, sympathic-adrenal-medullary system; SCN, suprachiasmatic nucleus; SNS, sympathetic nervous system; VLM, ventrolateral medulla. Core areas involved in emotional modulation are depicted in black. The SNS originates in brainstem nuclei and through preganglionic acetylcholinenergic (Ach) efferent fibers projects to postganglionic sympathetic ganglia. The long postganglionic neurons terminate on effector tissues, releasing primarily norepinephrine. Preganglionic neurons can also directly synapse with the postganglionic chromaffin cells of the adrenal medulla. Respectively, sympathetic activation depends on two peripheral branches (neural and adrenal) and releases norepinephrine (primarily locally) or adrenaline (systematically from the adrenal medulla). On the other hand, PNS activity is only displayed by nerves, through the comparatively long preganglionic ACh-neurons arising from brainstem and spinal sacral region (S2–S4) nuclei, which synapse with short postganglionic neurons within terminal ganglia close to or embedded to effector tissues. The preganglionic neurons that arise from the brainstem exit the CNS through the cranial nerves [N. occulomotorius (III); N. facialis (VII); N. glossopharyngeus (IX); N. vagus (X)]. The vagus nerve has major physiological significance, as about ¾ of all PNS fibers originate from the vagus. The parasympathetic response to stress is mainly mediated by the NAMB and the DMV, possibly through input from the NTS. Autonomic function is additionally modulated by respiratory function (i.e., respiratory sinus arrhythmia), blood pressure feedback systems (e.g., baroreflex circuit) mainly through the NTS, and by the endocrine systems (e.g., SAM, HPA axis). The hippocampus is only indirectly involved, e.g., in episodic (contextual fear) memory-related autonomic adjustments.

**Figure 2 biomedicines-11-01591-f002:**
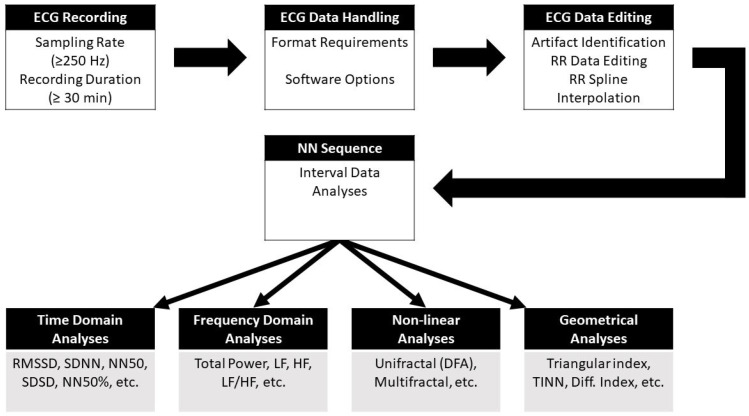
Schematic overview of the consecutive steps and processes involved from ECG recording to heart rate variability analyses in the different analytical domains. The abbreviations of the different measures are provided in Table 1. Modified from [45].

**Table 1 biomedicines-11-01591-t001:** Important heart rate measures in the four different analytical domains.

Variable	Units	Description
Time domain, statistical measures		
NN interval	ms	Normalized time between two heartbeat QRS events after excluding unreliable QRS intervals
SDNN	ms	Standard deviation of all NN intervals
SDANN	ms	Standard deviation of the averages of NN intervals in all 5 min segments of the recording
RMSSD	ms	The square root of the mean of differences between adjacent NN intervals
SDNN index	ms	Mean of the standard deviations of all NN intervals for all 5 min segments of the recording
SDSD	ms	Standard deviation of differences between adjacent NN intervals
NN50 count		Number of pairs of adjacent NN intervals differing by more than 50 ms in the entire recording
NN50	%	NN50 count divided by the total number of all NN intervals
Time domain, geometric measures		
Triangular index (HRV)		Total number of all NN intervals divided by the height of the histogram of all NN intervals measured on a discrete scale with bins of 1/128 s
TINN	ms	Baseline width of the minimum square difference triangular interpolation of the highest peak of the histogram of all NN intervals
Differential index	ms	Difference between the widths of the histogram of differences between adjacent NN intervals measured at selected heights
Logarithmic index	ms^−1^	Coefficient f of the exponential curve ke^−ϕt^, which is the best approximation of the histogram of absolute differences between adjacent NN intervals
Frequency domain, short-term recordings (5 min)		
Total power	ms^2^	Variance of all NN intervals (≈≤0.4 Hz)
VLF	ms^2^	Power in VLF range (f ≤ 0.04 Hz)
LF	ms^2^	Power in LF range (0.04 ≤ f ≤ 0.15 Hz)
LF norm		LF power in normalized units: LF/(Total power—VLF) × 100
HF	ms^2^	Power in HF range (0.15 ≤ f ≤ 0.4 Hz)
HF norm		HF power in normalized units: HF/(Total power—VLF) × 100
LF/HF		Ratio LF/HF—(Inaccurate) Index of sympatho-vagal balance
Frequency domain, long-term recordings (24 h)		
Total power	ms^2^	Variance of all NN intervals (≈≤0.4 Hz)
ULF	ms^2^	Power in the ULF range (f ≤ 0.003 Hz)
VLF	ms^2^	Power in the VLF range (0.003 ≤ f ≤ 0.04 Hz)
LF	ms^2^	Power in the LF range (0.04 ≤ f ≤ 0.15 Hz)
HF	ms^2^	Power in the HF range (0.15 ≤ f ≤ 0.4 Hz)
α		Slope of the linear interpolation of the spectrum in a log-log scale (f ≤ 0.01 Hz)
Nonlinear Domain—Unifractal		
α_fast_, α_slow_		Scaling coefficients of the detrended fluctuation analysis (DFA) with regard to short- (α_fast_) and long-term correlation (α_slow_)
Nonlinear Domain—Multifractal		
τ(q)		Multifractal spectrum

Modified from [45,46].

## Data Availability

Not applicable.

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
