# Peer review of "Heart Rate Variability as a Translational Dynamic Biomarker of Altered Autonomic Function in Health and Psychiatric Disease"

_biomedicines, 2023, doi:10.3390/biomedicines11061591_

Round 1

Reviewer 1 Report

The Authors submitted a Review Article aiming at highlighting the utility of heart rate variability (HRV) as a transdiagnostic dynamic biomarker for stress system sensitivity and vulnerability to stress-related disorders and support a much broader use in research and clinical practice.

The manuscript is well written and presented; the topic is also interesting. The English grammar and structure should be revised, while a double check for typos is also warranted. The abstract should be refined, giving a precise aim for this review and what will be discussed. A brief introduction is also recommended in which the latter purposes and the need for this review are introduced.

Author Response

We would like to thank the Reviewer for the detailed comments and suggestions, which helped to considerably improve the overall quality of the manuscript. We provide a detailed point-by-point response to all queries of the Reviewers and have included yellow highlighting in order to mark applied changes (additions) in the body of the revised manuscript.

Responses to Comments of Reviewer 1

Comment #1: The manuscript is well written and presented; the topic is also interesting.

Response: We appreciate these positive comments!

Comment #2: The English grammar and structure should be revised, while a double check for typos is also warranted.

Response: We acknowledge this point raised and have revised our manuscript accordingly.

Comment #3: The abstract should be refined, giving a precise aim for this review and what will be discussed.

Response: We have followed the Reviewer’s suggestion and now provide a clear aim and structure of this review and its contents in the abstract.

Comment #4: A brief introduction is also recommended in which the latter purposes and the need for this review are introduced.

Response: We acknowledge this point raised by the Reviewer and now provide a short introduction at the beginning of our manuscript accordingly (Section 1).

Reviewer 2 Report

This is an interesting and quite technical review on the how heart rate variability can be a biomarker of altered function of the autonomic nervous system in normal and psychiatric patients. 

I personally think that the review could be improved inclusing discussion of:

1) Specific examples and cases of psychiatric conditions in which HRV can be seen as a biomarker. So far this is discussed vaguely, some examples might render the text more interesting.

2) A more detailed analysis of the effects of beta-blockers and antiarrythmic agents such amiodarone on HRV, and how this could be taken in account in clinical conditions involving psychiatric conditions.

I would possibly change the fist five lines of the introduction as they are copied by "The neuroendocrinology of stress: the stress-related continuum of chronic disease development" in Mol Psy, but it is not really necessary.

I would also avoid the use of abbreviations when not absolutely necessary (e.g. HIP in line 148 is only used here in the text and the abbrevation is only seen in the legend of Fig. 1).

Author Response

We would like to thank the Reviewer for the detailed comments and suggestions, which helped to considerably improve the overall quality of the manuscript. We provide a detailed point-by-point response to all queries of the Reviewers and have included yellow highlighting in order to mark applied changes (additions) in the body of the revised manuscript.

Responses to Comments of Reviewer 2

Comment #1: This is an interesting and quite technical review on the how heart rate variability can be a biomarker of altered function of the autonomic nervous system in normal and psychiatric patients. 

Response: We appreciate these positive comments!

Comment #2: Specific examples and cases of psychiatric conditions in which HRV can be seen as a biomarker. So far this is discussed vaguely, some examples might render the text more interesting.

Response: We have followed the Reviewer’s suggestion and now provide examples of psychiatric conditions, where HRV has been used as a biomarker in the last paragraph of Section 6.

Comment #3: A more detailed analysis of the effects of beta-blockers and antiarrythmic agents such amiodarone on HRV, and how this could be taken in account in clinical conditions involving psychiatric conditions.

Response: We acknowledge this important issue raised and now provide additional examples of pharmacological effects on HRV in the 2nd and 3rd paragraph of Section 9 accordingly.

Comment #4: I would possibly change the first five lines of the introduction as they are copied by "The neuroendocrinology of stress: the stress-related continuum of chronic disease development" in Mol Psy, but it is not really necessary.

Response: We acknowledge this point raised by the Reviewer and have revised our manuscript accordingly.

Comment #5: I would also avoid the use of abbreviations when not absolutely necessary (e.g., HIP in line 148 is only used here in the text and the abbreviation is only seen in the legend of Fig. 1).

Response: We have followed the Reviewer’s suggestion and have revised our manuscript accordingly.
